# Sinonasal Stent Coated with Slow-Release Varnish of Chlorhexidine Has Sustained Protection against Bacterial Biofilm Growth in the Sinonasal Cavity: An In Vitro Study

**DOI:** 10.3390/pharmaceutics13111783

**Published:** 2021-10-25

**Authors:** Alessandra Cataldo Russomando, Ronit Vogt Sionov, Michael Friedman, Irith Gati, Ron Eliashar, Doron Steinberg, Menachem Gross

**Affiliations:** 1Department of Otolaryngology-Head and Neck Surgery, Hadassah Medical Center, Jerusalem 9112102, Israel; roneliashar@gmail.com (R.E.); drgrossm@hotmail.com (M.G.); 2The Biofilm Research Laboratory, The Institute of Dental Sciences, The Faculty of Dental Medicine, The Hebrew University of Jerusalem, Jerusalem 9112102, Israel; ronit.sionov@mail.huji.ac.il (R.V.S.); dorons@ekmd.huji.ac.il (D.S.); 3Institute for Drug Research, School of Pharmacy, The Hebrew University of Jerusalem, Jerusalem 9112102, Israel; michaelf@ekmd.huji.ac.il (M.F.); irith.gati@mail.huji.ac.il (I.G.); 4Faculty of Medicine, The Hebrew University of Jerusalem, Jerusalem 9112102, Israel

**Keywords:** bacteria biofilm, biofilm inhibition, chlorhexidine, chronic rhinosinusitis, nasal stent, *Pseudomonas aeruginosa*, *Staphylococcus aureus*, sustained release varnish (SRV)

## Abstract

The aim of the study was to develop a sustained-release varnish (SRV) containing chlorhexidine (CHX) for sinonasal stents (SNS) to reduce bacterial growth and biofilm formation in the sinonasal cavity. Segments of SNS were coated with SRV-CHX or SRV-placebo and exposed daily to bacterial cultures of *Staphylococcus aureus subsp. aureus* ATCC 25923 or *Pseudomonas aeruginosa* ATCC HER-1018 (PAO1). Anti-bacterial effects were assessed by disc diffusion assay and planktonic-based activity assay. Biofilm formation on the coated stents was visualized by confocal laser scanning microscopy (CLSM) and high-resolution scanning electron microscopy (HR-SEM). The metabolic activity of the biofilms was determined using the 3-(4,5-dimethyl-2-thiazolyl)-2,5-diphenyl-2H-tetrazolium bromide (MTT) method. Disc diffusion assay showed that SRV-CHX-coated SNS segments inhibited bacterial growth of *S. aureus*
*subsp. aureus* ATCC 25923 for 26 days and *P. aeruginosa* ATCC HER-1018 for 19 days. CHX was released from coated SNS segments in a pH 6 medium up to 30 days, resulting in growth inhibition of *S. aureus*
*subsp. aureus* ATCC 25923 for 22 days and *P. aeruginosa* ATCC HER-1018 for 24 days. The MTT assay showed a reduction of biofilm growth on the coated SNS by 69% for *S. aureus*
*subsp. aureus* ATCC 25923 and 40% for *P. aeruginosa* ATCC HER-1018 compared to the placebo stent after repeated exposure to planktonic growing bacteria. CLSM and HR-SEM showed a significant reduction of biofilm formation on the SRV-CHX-coated SNS segments. Coating of SNS with SRV-CHX maintains a sustained delivery of CHX, providing an inhibitory effect on the bacterial growth of *S. aureus*
*subsp. aureus* ATCC 25923 and *P. aeruginosa* ATCC HER-1018 for approximately 3 weeks.

## 1. Introduction

Chronic rhinosinusitis (CRS) is a clinical manifestation of mucosal inflammation of the paranasal sinuses that has lasted more than 12 weeks, leading to several cardinal sinonasal symptoms including nasal congestion, nasal drainage, facial pain, and anosmia [1]. CRS is a significant health problem and affects 5–12% of the general population [1], with estimated total costs that exceed USD 30 billion per year in the USA, with USD 20 billion accounted for indirect costs primarily due to lost productivity in those suffering from rhinosinusitis [2].

The causes of CRS are multifactorial. Microbial biofilm formation in the sinonasal cavity plays a pathogenic role in 29 to 72% of the cases [3]. Microbiota involvement results in increased disease severity, with patients showing more severe objective clinical disease indicators [4,5] and a higher burden of subjective symptoms [5,6]. Biofilm-positive CRS patients demonstrate significantly worse objective outcomes with a more severe disease on pre- and post-operative nasal endoscopy, more follow-up visits, and additional courses of antibiotics when compared to biofilm negative CRS patients [3]. The presence of biofilms in CRS is also independently associated with a greater risk of recurrence [5] and revision surgery [7].

Biofilms are organized communities of bacteria, enwrapped in a self-produced exopolymeric matrix of polysaccharides, proteins, and nucleic acids, which form a protective matrix around the microorganisms [8]. The biofilms frequently show resistance to host defense mechanisms and antibiotic therapy [9]. Multiple microorganisms have been implicated in these biofilms, including *S. aureus*, *P. aeruginosa*, *Haemophilus influenza,* and *Moraxella cattarhalis* [10,11]. Among these, *S. aureus* and *P. aeruginosa* biofilms are strongly associated with severe, recurrent, and recalcitrant cases of CRS [12,13,14].

Prolonged use of antibiotic treatment often alleviates symptoms during CRS exacerbations but may fail to eradicate the biofilm nidus, resulting in a relapsing and remitting disease course [15]. Treatment of refractory biofilm-associated CRS is multifactorial, including systemic antibiotics, local therapy, and surgery. However, various antibiotic resistant mechanisms developed in up to 46% of the patients [15], urging the need for a long-term solution to prevent biofilm formation in the nasal cavity. Szaleniec et al. [15] suggested treating the refractory CRS patients with phage therapy, while Ezzat et al. [16] tried a topical application of ofloxacin eye drops. Both approaches showed a certain level of success.

Herein, we propose a different method of eliminating and preventing biofilm accumulation by coating a standard sinonasal stent (SNS) with a sustained release varnish (SRV) containing an anti-microbial drug, prolonging the duration of the drug release at the target site. Advantages of this approach include the local release of the drug in therapeutic active doses for long periods while simultaneously resulting in less toxic adverse effects since the concentration is much lower than the drug delivery by sprays and gels. SRV embedded with different active agents has previously been shown to provide long-term antimicrobial activities in other medical systems including oral care [17] and catheter-associated urinary tract infections [18]. Chlorhexidine (CHX), which is selected as the active agent in this study, is a quaternary ammonium antiseptic compound, with known safety and effectivity in human mucosal membranes and has previously been tested for medical device use, including the nasopharynx [17,19,20]. In addition, CHX has broad-spectrum antimicrobial activity against both Gram-positive and Gram-negative bacteria as well as yeasts [21,22,23,24] and has been shown to act on preformed biofilms [23]. 

SNS is used for the prevention of lateralization of the middle concha and to liberate steroids preventing the recurrence of nasal polyp [25]. Currently, no commercially available stents can reduce biofilms in the nasal cavity. The present study aimed to evaluate the antibacterial and anti-biofilm effects of a sustained release medicated varnish on sinonasal stents using the two pathogens *S. aureus*
*subsp. aureus* ATCC 25923 and *P. aeruginosa* ATCC HER-1018 (PAO1) in an in vitro model as a step towards future in vivo trials. Although a similar slow-release system has been shown to be efficient in preventing biofilm formation on siliconized latex Foley catheters [18], it was important to study whether this system can also be used to coat nasal stents that are composed of polyurethane plastic and nitinol wires.

## 2. Material and Methods

### 2.1. Sustained Release Varnish (SRV) Preparation

The SRV containing chlorhexidine (CHX) was formulated as described [18] with slight modification. The SRV-CHX varnish contained 2% (*W*/*W*) CHX (Sigma, St. Louis, MO, USA), 4.5% ethylcellulose (Ethocel N-100, Hercules Inc., Wilmington, DE, USA), 0.6% polyethylene glycol 400 (PEG400; Sigma) and 0.5% hydroxypropyl cellulose (Klucel EF, Ashland Specialty Ingredients, Switzerland) in ethanol, resulting in the formation of a dry film containing 26.3% CHX (*W*/*W*). The placebo varnish (SRV-placebo) was prepared identically to SRV-CHX but omitting the CHX from the formulation.

### 2.2. Coating of the Stents

The sinus-nasal stent (SNS) ArchSinus (S.T.S Medical LTD, Misgav, Israel) were cut into 1 cm segments, disinfected with 70% ethanol, and triple-coated with SRV-CHX or SRV-placebo, resulting in 40–50 mg films per segment equivalent to 10.5–13.1 mg CHX. The coating was performed by immersing the SNS pieces in the SRV and let dry at room temperature until a film was formed on the surface of the pieces. This process was repeated twice. The pieces were then allowed to dry completely before use. The effect of SRV-CHX was compared to that of SRV-placebo.

### 2.3. Bacterial Strains

A frozen stock of *S. aureus*
*subsp. aureus* ATCC 25923 or *P. aeruginosa* ATCC HER-1018 (PAO1) were daily inoculated in tryptic soy broth (TSB) (Acumedia, Neogen, Lansing, MI, USA) at a ratio of 1:100 and incubated overnight at 37 °C until reaching an OD_600nm_ of 1.8–2.0. The overnight bacterial culture was diluted in TSB containing 1% D-glucose (TSBG) and used for the biological assays described below.

### 2.4. Agar Diffusion Sensitivity Assay

The coated SNS segments were repeatedly placed on tryptic soy agar (TSA) plates pre-seeded with 100 µL of an overnight culture of either *S. aureus*
*subsp. aureus* ATCC 25923 or *P. aeruginosa* ATCC HER-1018 (PAO1) and incubated at 37 °C for 24 h. The zone of inhibition around the placed SNS was measured daily, and the SNS segments were transferred to new bacteria-seeded agar plates for further incubation. The zone of inhibition was calculated according to the formula: (d1/2) * (d2/2) * π, where d1 and d2 are the two diameters of the clearance zone.

### 2.5. Planktonic Bacterial Growth and Biofilm Formation

SRV-CHX and SRV-placebo-coated stents were repeatedly incubated in 1 mL of TSBG medium with 10 µL of an overnight bacterial culture for 24 h for a total of 7 days. The OD at 600 nm was measured daily using the Ultraspec 10 Spectrophotometer (Amersham Biosciences, Buckinghamshire, UK). TSBG medium incubated with the stents without bacteria served as blank. The extent of biofilm formation on the stents was analyzed by different techniques as described below.

### 2.6. Evaluation of Biofilm Development by Confocal Laser Scanning Microscopy (CLSM)

The SNS segments were stained with 3.3 µM SYTO 9 (Invitrogen, Life Technologies, Eugene, OR, USA) and 10 µM propidium iodide (PI) (Sigma) in PBS for 30 min [26] and visualized by a spinning disk confocal microscope (Nikon Yokogawa W1 Spinning Disk, Tokyo, Japan, with 50 µm pinholes) or a Nikon eclipse Ti-U confocal microscope (Nikon Instruments, Tokyo, Japan). The biofilm depth was assessed on spinning disk microscope by capturing optical cross-sections at 2.5 μm intervals from the bottom of the biofilm to its top. The SYTO 9 green fluorescence dye, which enters both live and dead bacteria, was visualized using 488 nm excitation and 515 nm emission filters. The PI red fluorescence dye, which only penetrates dead bacteria, was measured using 543 nm excitation and 570 nm emission filters. Thus, live bacteria fluoresce green light, while dead bacteria fluoresce both green and red light. Three-dimensional images of the formed biofilms were reconstructed using the NIS-Element AR software. This software was also used to analyze the fluorescence intensity of SYTO 9 and PI staining in each captured layer of the biofilms. The percentage of total biomass of viable cells in biofilm formed in the presence of SRV-CHX was calculated in comparison to SRV-placebo.

### 2.7. High-Resolution Scanning Electron Microscopy (HR-SEM)

The SNS segments were fixed with 4% paraformaldehyde in double-distilled water (DDW) for 20 min, washed with DDW, and dried before coating with iridium and visualized using an FEI Magellan 400L HR-SEM (FEI Company, Hillsboro, OR, USA) at ×200–×5000 magnifications.

### 2.8. Evaluation of Biofilm Metabolic Activity

The metabolic activity of biofilms on the stents was determined by the MTT method [27]. The washed stents were exposed to 0.5 mg/mL MTT (3-(4,5-dimethyl-2-thiazolyl)-2,5-diphenyl-2H-tetrazolium bromide; Sigma, St. Louis, MO, USA) in PBS for 1 h, washed, and the amount of tetrazolium formed was determined at an OD of 570 nm after dissolving it in 500 μL DMSO.

### 2.9. Determination of CHX Release from Coated SNS at pH 6

Coated stents were repeatedly incubated in 1 mL of PBS adjusted to pH 6.0 for 24 h for a total of 30 days. The CHX concentration was determined spectrophotometrically by measuring the OD at 255 nm in a Nanodrop spectrophotometer (NanoDrop One Thermo Fisher Scientific, Waltham, MA, USA) and calculated against a standard curve made by known concentrations of CHX. The anti-bacterial effect of the released CHX was tested by placing 20 µL of the daily collected fluids on agar plates seeded with *S. aureus*
*subsp. aureus* ATCC 25923 or *P. aeruginosa* ATCC HER-1018, followed by a 24 h incubation at 37 °C [27]. Zones of inhibition were measured daily and presented as the area of clearance (cm^2^).

### 2.10. Statistical Analysis

The data are presented as the average ± standard deviation from 3–4 independent experiments, using a 1 cm coated stent in each experiment. Statistical analysis was performed using the Microsoft Excel software. Student’s *t*-test was used to compare CHX-coated stents with placebo-coated stents, with a *p*-value less than 0.05 considered significant.

## 3. Results

We used a varnish containing chlorhexidine (CHX) and biocompatible polymers that form a film with sustained-release properties upon evaporation of ethanol [17]. The coating can be done on the closed stent, and the varnish is still firmly adhered to the stent upon its opening (Figure 1) which is required for better positioning of the stent in the nasal cavity. Both the plastic and metallic parts were coated by the varnish. Anti-bacterial and anti-biofilm effects of SNS coated with SRV-CHX in comparison to SRV-placebo were tested on *S. aureus*
*subsp. aureus* ATCC 25923 and *P. aeruginosa* ATCC HER-1018 using agar diffusion assay, planktonic growth, CLSM, HR-SEM, and metabolic assay.

### 3.1. Agar Disk Diffusion Assay Shows the Prolonged Effect of SRV-CHX on Bacterial Clearance 

Initially, we wanted to test for how long time we could achieve an anti-bacterial effect of nasal stents coated with SRV-CHX. To this end, the SRV-CHX or SRV-placebo coated stents were daily placed on agar plates pre-seeded with either *S. aureus*
*subsp. aureus* ATCC 25923 or *P. aeruginosa* ATCC HER-1018 and the clearance zone determined after a 24 h incubation (Figure 2A). The time-course experiments showed that SRV-CHX-coated SNS segments inhibited bacterial growth of *S. aureus*
*subsp. aureus* ATCC 25923 for 26 ± 4 days and *P. aeruginosa* ATCC HER-1018 for 19 ± 5 days (Figure 2B,C). SRV-placebo did not show any antibacterial effect (Figure 2A). The clearance zone was higher during the first period of 15 days, but still sufficient amount of CHX was released during the later period to cause the elimination of the surrounding bacteria. Some fluctuations in the size of the clearance zone were observed during the experimental period that can be explained by some variance in the amount of CHX released, as will be described below.

### 3.2. Antibacterial Effect of Coated Stent on Planktonic Bacterial Growth

Next, we studied the effect of SRV-CHX- and SRV-placebo-coated SNS segments on planktonic bacterial growth. To this end, the coated stents were daily incubated with fresh cultures of either *S. aureus*
*subsp. aureus* ATCC 25923 or *P. aeruginosa* ATCC HER-1018, and the OD at 600 nm was measured after each incubation of 24 h. SNS-coated with SRV-CHX caused a significant decrease in the bacterial growth of both *S. aureus*
*subsp. aureus* ATCC 25923 (90 ± 6% reduction) and *P. aeruginosa* ATCC HER-1018 (89 ± 4% reduction) when compared to SRV-placebo-coated SNS at day 7 (Figure 3A). High OD in the SRV-CHX-coated stents in the presence of the two bacterial species during the initial four days is likely due to initial high CHX release that precipitates out proteins in the growth medium, causing turbidity. To test the viability of the bacteria in these samples, three drops of 10 µL of each sample were seeded on agar plates, and the bacterial growth was inspected after a 24 h incubation. This assay showed no bacterial growth from cultures exposed to SRV-CHX during the entire test period of 7 days, while bacterial growth was seen, as expected, in the cultures exposed to SRV-placebo (Figure 3B), indicating that the amount of CHX released was sufficient to prevent bacterial growth in its surroundings.

### 3.3. Anti-Bacterial Activity of CHX Released from Coated Stent at pH 6

Sinonasal cavity pH tends to be slightly acidic [28], as such, the time-course release was performed in a solution of pH 6, to evaluate the resultant antibacterial activity towards *S. aureus*
*subsp. aureus* ATCC 25923 and *P. aeruginosa* ATCC HER-1018. A significantly higher CHX release rate was recorded during the first day (300 µg/mL (0.03%) CHX, corresponding to 8.1% of the total CHX content of the coated film) in comparison to the next 19 days, where 100–200 µg/mL (0.01–0.02%) CHX was released daily (Figure 4A). Up to day 20, 87% of the total CHX content was released from the coated film (Figure 4B). Thereafter, the daily CHX release declined from 87 µg/mL (0.0087%) at day 21 to 14 µg/mL (0.0014%) at day 30 (Figure 4A). The amount of CHX released into the pH 6 medium was sufficient to inhibit the growth of *S. aureus*
*subsp. aureus* ATCC 25923 for 21 days and *P. aeruginosa* ATCC HER-1018 for 24 days (Figure 4C,D). Some fluctuations in the size of the clearance zone were observed, likely due to experimental variance. Nevertheless, the amount of CHX released during the stated periods was sufficient to prevent bacteria growth. Furthermore, the data demonstrate a minimum concentration of CHX of 0.005–0.01% is necessary to achieve growth inhibition.

### 3.4. Reduced Biofilm Formation on SRV-CHX-Coated SNS 

After 7 days incubation of the coated SNS segments with planktonic growing *S. aureus*
*subsp. aureus* ATCC 25923 or *P. aeruginosa* ATCC HER-1018, the biofilm mass formed on the SNS was studied by metabolic MTT assay, dead/live SYTO 9/PI staining visualized by CLSM, and HR-SEM. The MTT assay showed that the biofilm formation by *S. aureus*
*subsp. aureus* ATCC 25923 on the SRV-CHX stent was reduced by 69%, and the biofilm formation by *P. aeruginosa* ATCC HER-1018 on the SRV-CHX stent was reduced by 40% in comparison to SRV-placebo.

CLSM images of SRV-placebo samples exposed for 7 days to planktonic *S. aureus*
*subsp. aureus* ATCC 25923 showed a strong SYTO 9 (green) fluorescence staining with only scattered PI (red) fluorescence (Figure 5A), indicating a continuous biofilm layer of mostly live bacteria. In contrast, the SRV-CHX-coated stent showed scattered green fluorescence together with red fluorescence, indicating most bacteria were dead (Figure 5B). The thickness of the *S. aureus*
*subsp. aureus* ATCC 25923 biofilm was reduced by 76.6% on the SRV-CHX-coated stent compared to the SRV-placebo-coated stent (Figure 5C,D). Quantifying the relative green and red fluorescence of the biofilms (Figure 5C,D), we observed that 2.4% of the *S. aureus*
*subsp. aureus* ATCC 25923 on the stent with SRV-placebo were dead, while 95% of the *S. aureus*
*subsp. aureus* ATCC 25923 on the stent with SRV-CHX were dead. Likewise, CLSM images of the SRV-placebo samples exposed for 7 days to planktonic growing *P. aeruginosa* ATCC HER-1018 showed green fluorescence with almost no red fluorescence, indicating that the stent is covered by a biofilm of predominantly live bacteria (Figure 6A–C). On the contrary, the SRV-CHX-coated stent shows both green and red fluorescence (Figure 6D–F), implying a high fraction of dead bacteria. Quantification of the relative green and red fluorescence of the biofilms showed that 2.5% of the *P. aeruginosa* ATCC HER-1018 on the SRV-placebo-coated stent were dead bacteria, in contrast to 66% dead bacteria on the SRV-CHX-coated stent.

HR-SEM images demonstrated a strong inhibition of biofilm formation on the SNS coated with SRV-CHX for both bacterial strains in comparison to SRV-placebo-coated segments (Figure 7). Continuous biofilms of both *S. aureus*
*subsp. aureus* ATCC 25923 and *P. aeruginosa* ATCC HER-1018 were manifested on the SRV-placebo-coated stents (Figure 7A,B), whereas only scattered clusters of *S. aureus*
*subsp. aureus* ATCC 25923 and scattered single cells of *P. aeruginosa* ATCC HER-1018 were observed on SRV-CHX-coated stents (Figure 7C,D). While extracellular biofilm matrix was observed in the biofilms formed on the stent coated with SRV-placebo (Figure 7A,B), there was no sign of extracellular matrix on the stents coated with SRV-CHX (Figure 7C,D). Altogether, these data demonstrate that coating the stent with SRV-CHX significantly reduces biofilm formation.

## 4. Discussion

Effective surgical management of medically refractory CRS relies on the establishment of a patent and durable opening into the paranasal sinuses, to allow for the efficient drainage of sinus secretion and delivery of topical therapies. SNSs are frequently inserted into the nasal cavity following functional endoscopic sinus surgery (FESS) to prevent restenosis and scarring of the neo-ostium, helping to keep the sinus outflow tract.

Biofilm-positive CRS patients demonstrate significantly worse objective outcomes with a more severe disease on pre-and post-operative nasal endoscopy, more follow-up visits, and additional courses of antibiotics when compared to biofilm negative CRS patients [3]. We proposed that coating these stents with a slow-release varnish (SRV) that contains the antiseptic chlorhexidine (CHX), would be useful for eradicating biofilms in the nasal cavity and preventing biofilm formation on the stents. It was important to study whether the coating could be retained on the stents for long periods of time, and concomitant release a sufficient amount of CHX to the surroundings to prevent bacterial growth.

To the best of our knowledge, this is the first in vitro study applying chlorhexidine in a sustained release technology as a coating on nasal stents for the prevention of biofilm formation in the nasal cavity. CHX is one of the most commonly prescribed antiseptic agents in the dental field. It has a long-lasting antibacterial and antifungal activity with a broad spectrum of action [21,22,23,24], and its use is considered a safe compound, with minimal and transitory local and systemic side effects [29].

With their high tendency of biofilm formation, *S. aureus* and *P. aeruginosa* are frequently associated with poor clinical outcomes in individuals with CRS with or without nasal polyposis. The biofilm-producing tendency of *S. aureus* and *P. aeruginosa* affects the clinical outcome and may help explain the persistent disease in CRS [30].

SRV technology has numerous pharmacologic advantages stemming from the ability to slowly release the drug from the matrix. Specific advantages include prolonged duration of the drug at the desired site, better penetration into the biofilm layers, and reduced systemic and local side-effects as the amounts of drug released are low and locally targeted [31]. In addition, the sustained release local delivery system does not require repeated medication, such that patient compliance is not an obstacle in the healing process.

Drug release from the SRV matrix depends on several parameters including the type of polymers used, pharmaceutical additives, the solvent, the drug-loaded, the environment it is placed in, and the support material which is coated.

In our experiments, SNS coated with SRV-CHX showed a significant daily release of the active agent during the first 20–25 days that was sufficient to inhibit the growth of both *S. aureus subsp. aureus* ATCC 25923 and *P. aeruginosa* ATCC HER-1018. Thereafter, CHX continued to be released for another 6–7 days, ultimately approaching an amount of CHX that was insufficient to prevent bacterial growth. Sustained release of CHX was observed both in bacterial growth medium and medium with a pH of 6 that mimics the pH of the nasal mucosa that oscillates between 5.5–6.5. Çankaya et al. [32] compared the effect of various concentrations of CHX on the nasal mucosa of rabbits by spraying 150 μL of a CHX solution twice a day into the nasal cavity for 5 days. They observed that rabbits receiving 0.12% or 0.20% CHX showed high neutrophil infiltration in submucosa and loss of cilia. However, rabbits receiving 0.03% CHX showed only mild cell infiltration of the submucosa and the cilia appeared in normal form. Thus, our sustained-release film providing a concentration released of CHX of 0.03% and below should be safe; however, further studies are required to prove its safety in vivo.

It was important not only to prevent bacterial growth in the surroundings of the SNS but also to prevent biofilm formation on the stents. Therefore, we studied the biofilm formation on SRV-coated SNS segments after repeated exposure to planktonic growing *S. aureus*
*subsp. aureus* ATCC 25923 or *P. aeruginosa* ATCC HER-1018. We observed that the biofilm mass was significantly reduced on SRV-CHX-coated SNS when compared to SRV-placebo. The reduction in biofilm mass was especially notable when visualizing the bacteria on the coated stents using HR-SEM. MTT metabolic assay and CLSM imaging of live/dead stained stents also showed a tendency of SRV-CHX to reduce biofilm formation. It should be noted that each assay monitors different parameters. While HR-SEM directly visualizes the bacteria, the MTT assay measures the metabolic activity of the bacteria. In this respect, it needs to be considered that the bacteria enwrapped in the biofilm are often sessile, which contrasts with the metabolically active free-living bacteria. This can explain why we observed a less reduction in biofilm mass using the MTT assay in comparison to HR-SEM.

CLSM staining can distinguish between live and dead bacteria, where SYTO 9 enters both live and dead cells, whereas PI only the dead cells. Using this technique, most of the bacteria in the biofilms of SRV-placebo-coated SNS were alive, whereas a high percentage of the bacteria that have managed to adhere to the SRV-CHX coating was dead. The latter is not surprising, considering the anti-bacterial action of CHX so that even when being incorporated into a polymeric film, CHX exerts anti-bacterial activities.

It is important to mention that CHX has been reported to be safe in the nasopharynx as a gel [19], thus can be used as the active component in the SRV. Due to CHX being an antiseptic agent, it should minimize the risk of bacterial resistance. One of the limitations of this study is that we studied the anti-bacterial effect of SRV-CHX in an in vitro model using only two of the most common biofilm-related rhinosinusitis pathogens, while there are also other pathogens involved in CRS. Moreover, the efficacy of the SRV-CHX system on clinical mixed bacterial and fungal biofilms should be the aim for further studies. Notably, the SRV-CHX was developed as a pharmaceutical platform, and our in vitro study proved its efficacy, at least against *S. aureus*
*subsp. aureus* ATCC 25923 and *P. aeruginosa* ATCC HER-1018. 

Other antimicrobial agents or anti-inflammatory agents (e.g., steroids) or a combination of them may be used for the formulation of new SRVs. In this study, we have proven the concept that SRV technology can be used in in vitro models. Based on these results, in vivo clinical trials are indicated to further evaluate the in vivo efficiency of this technique.

## 5. Conclusions

This is a first attempt to develop an SRV drug delivery to the nose that intends to prevent biofilm formation of pathogenic bacteria. The SNS maintains the coating and sustains the delivery of CHX for around 20–24 days, which provided an inhibitory effect on biofilm formation by *S. aureus*
*subsp. aureus* ATCC 25923 and *P. aeruginosa* ATCC HER-1018. These promising results may play a significant role in the future treatment of chronic rhinosinusitis, reducing the need for revision endoscopic sinus surgery in patients with chronic sinusitis. Further studies evaluating the efficacy of SNS SRV-CHX in pre-clinical models are needed. 

## Figures and Tables

**Figure 1 pharmaceutics-13-01783-f001:**
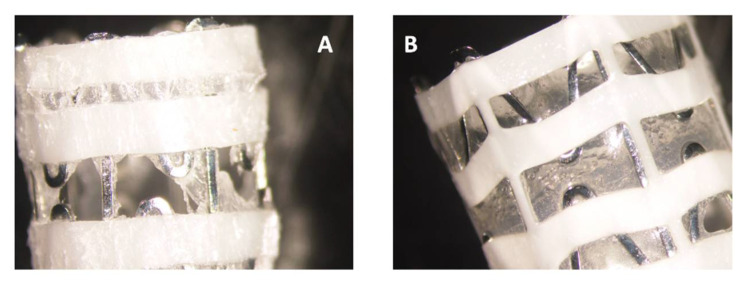
(**A**) SRV-CHX-coated stent in closed state. (**B**) The coating after opening the stent. The coating was retained on both polyurethane plastic and nitinol wires.

**Figure 2 pharmaceutics-13-01783-f002:**
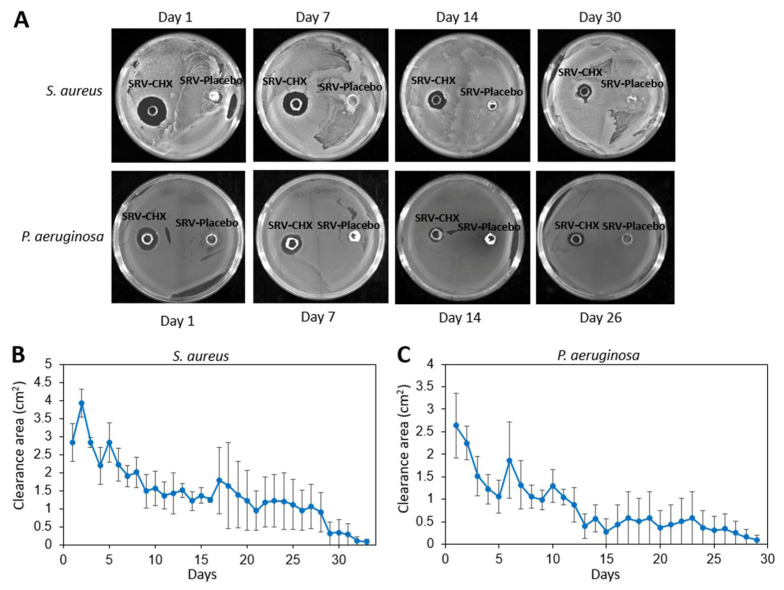
(**A**) Bacterial clearance around SRV-CHX-coated stents plated on fresh *S. aureus*
*subsp. aureus* ATCC 25923 (upper row) or *P. aeruginosa* ATCC HER-1018 (PAO1) (lower row) agar plates. The SRV-placebo-coated stents showed no clearance of bacteria. (**B**,**C**) The average bacterial inhibition zones were observed after daily transfer of SRV-CHX and SRV-placebo-coated stents on fresh *S. aureus*
*subsp. aureus* ATCC 25923 (**B**) or *P. aeruginosa* ATCC HER-1018 (**C**) agar plates. *n* = 3. *p* < 0.05 for SRV-CHX-coated stents versus SRV-placebo-coated stents.

**Figure 3 pharmaceutics-13-01783-f003:**
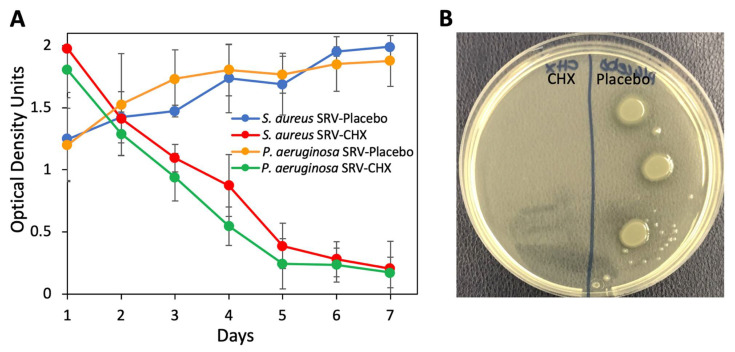
(**A**) SRV-CHX- and SRV-placebo-coated stents were incubated daily with planktonic growing bacteria, and the OD was measured after 24 h. *n* = 3. *p* < 0.05 SRV-CHX-coated stents versus SRV-placebo-coated stents. (**B**) Three drops of 10 µL of each sample at day 1 were seeded on agar plates followed by a 24 h incubation.

**Figure 4 pharmaceutics-13-01783-f004:**
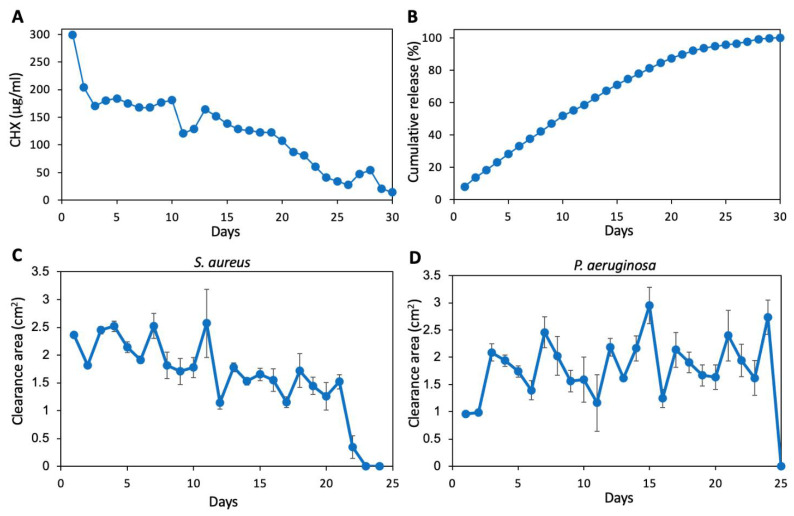
(**A**) Daily release of CHX from SRV-CHX-coated stent into pH 6 medium. *n* = 1. (**B**) Cumulative release of CHX into the medium. *n* = 1. (**C**,**D**) The clearance area of released CHX on *S. aureus*
*subsp. aureus* ATCC 25923 (**C**) or *P. aeruginosa* ATCC HER-1018 (**D**) plates. *n* = 4.

**Figure 5 pharmaceutics-13-01783-f005:**
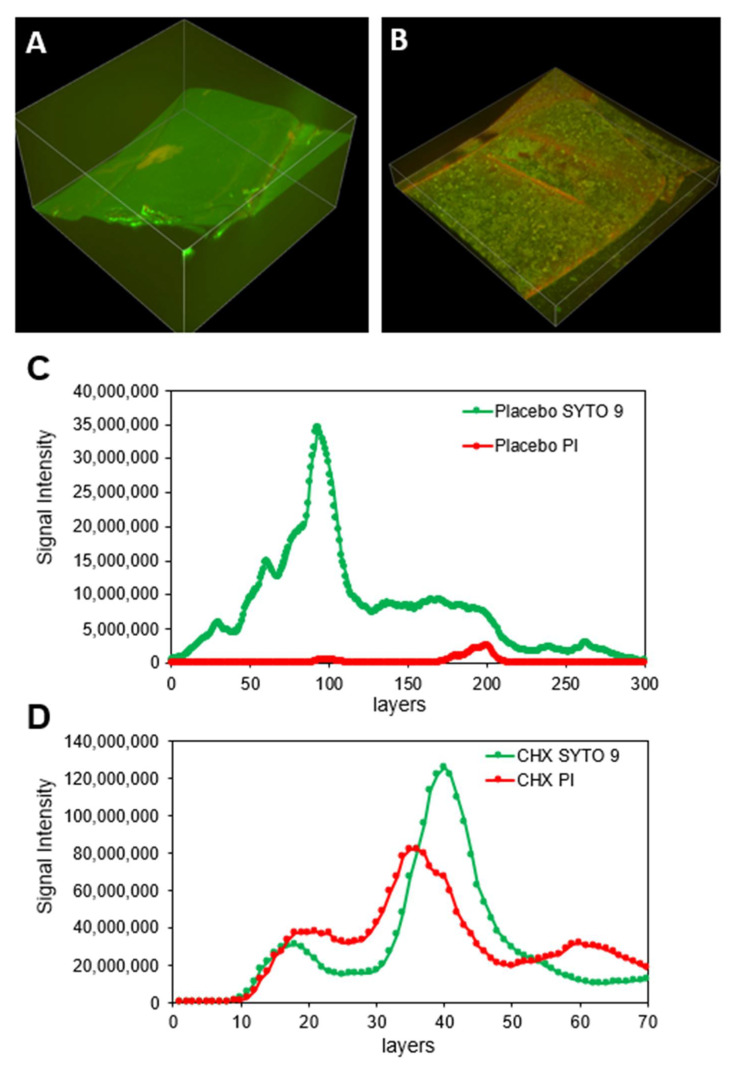
(**A**,**B**) SYTO 9/PI staining of *S. aureus*
*subsp. aureus* ATCC 25923 biofilm formed on SRV-placebo (**A**) and SRV-CHX (**B**)-coated SNS segments at day 7. (**C**,**D**) Quantification of the staining of stents coated with either SRV-placebo (**C**) or SRV-CHX (**D**).

**Figure 6 pharmaceutics-13-01783-f006:**
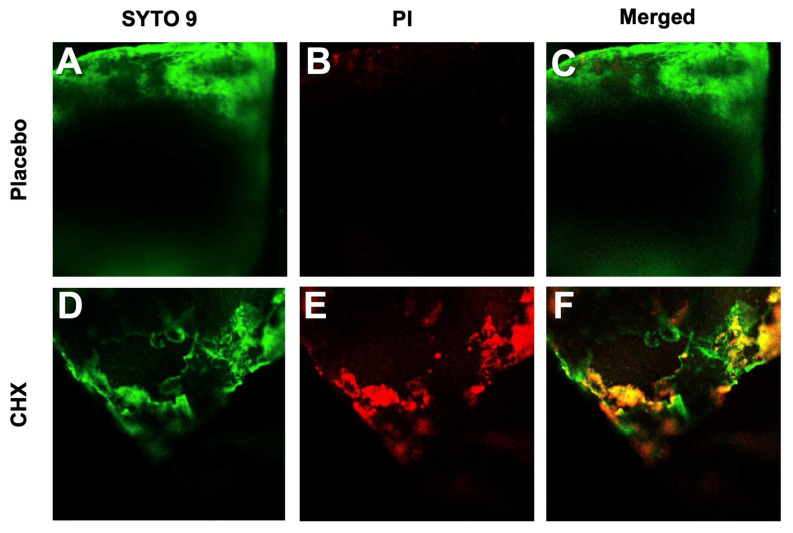
(**A**–**E**) SYTO 9/PI staining of *P. aeruginosa* ATCC HER-1018 biofilm formed on SRV-placebo- (**A**–**C**) and SRV-CHX-coated (**D**–**F**) SNS segments at day 7. The SNS segments were visualized by Nikon eclipse Ti-U confocal microscope.

**Figure 7 pharmaceutics-13-01783-f007:**
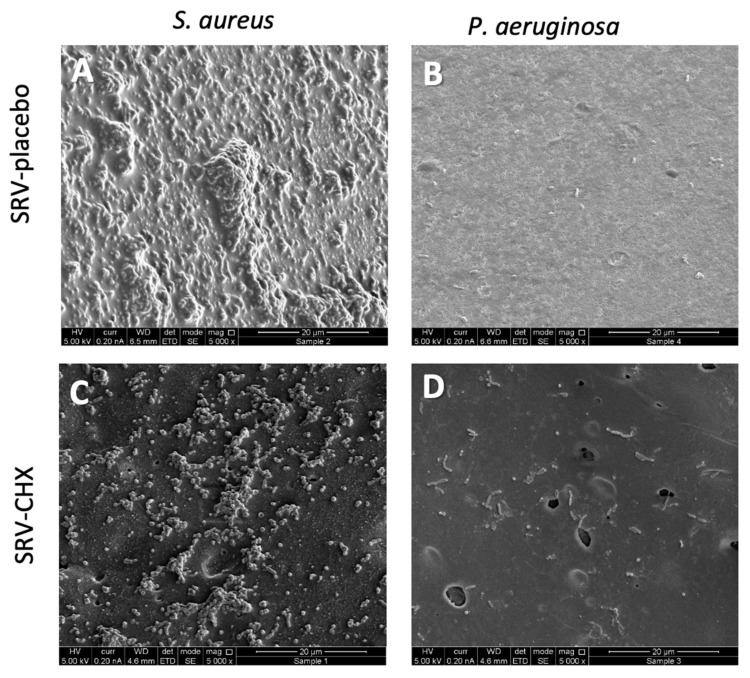
HR-SEM images of SRV-placebo- (**A**,**B**) and SRV-CHX-coated (**C**,**D**) SNS surfaces that have been exposed 7 times to planktonic growing *S. aureus subsp. aureus* ATCC 25923 (**A**,**C**) or *P. aeruginosa* ATCC HER-1018 (**B**,**D**). The ×5000 magnification is shown.

## Data Availability

Raw data are available upon reasonable request.

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
