# Peer review of "Sinonasal Stent Coated with Slow-Release Varnish of Chlorhexidine Has Sustained Protection against Bacterial Biofilm Growth in the Sinonasal Cavity: An In Vitro Study"

_pharmaceutics, 2021, doi:10.3390/pharmaceutics13111783_

Round 1

Reviewer 1 Report

Here I present the review of the paper entitled “Sinonasal stent coated with slow-release varnish of chlorhexidine has sustained protection against bacterial biofilm growth in the sinonasal cavity: An in vitro study” submitted to Pharmaceuticals.

Paper describes manufacturing of sinonasal stents coated with chlorhexidine as well as asses the antibacterial properties of produced material.  

Novelty of the paper is questionable. Study by Zelichenko et al., described SRV-CHX stents for urethra (Zelichenko, Genady, et al. "Prevention of initial biofilm formation on ureteral stents using a sustained releasing varnish containing chlorhexidine: in vitro study." Journal of endourology 27.3 (2013): 333-337.) Moreover, the authors previously described method of SRV-CHX stent production (Gefter Shenderovich, J.; Zaks, B.; Kirmayer, D.; Lavy, E.; Steinberg, D.; Friedman, M., Chlorhexidine sustained-releasevarnishes for catheter coating - Dissolution kinetics and antibiofilm properties. Eur J Pharm Sci 2018, 112, 1-7. ) What is actually novel in your study?

Abstract properly covers the main points of the study and key words are properly chosen. Title is clear and effective. Authors cited 20 references, majority of them were published within last 10 years. Study design does not rise ethical concerns. Language quality is sufficient

Critical issues

  • Please provide more data on antimicrobial properties of stents. Usage of only two bacterial species is not enough. Please report antibacterial properties of your stents against other pathogens associated with rhinosinusitis.
  • Usage of only two bacterial strains is not enough. It is advised to use more than one strain of each pathogen. Typically, bacterial strains have different susceptibility to xenobiotics.
  • Authors used only reference strains of bacteria. Lack of assessment of antimicrobial properties of patient-isolated bacteria is another downfall of the paper.
  • Study lacks any data about cytotoxicity of the stents. It is impossible to analyze antimicrobial properties of stents without knowing their impact on mammalian cells. In my opinion at least simple cytotoxicity screen (e.g. MTT, XTT or almar blue test) should be performed with usage of at least two cell lines derived from nasopharynges.

Major issues

  • Introduction is vaguely written. Please proved more detail on the diseased as well as current knowledge about SRV stents.
  • Discussion is way too short. Please discuss your finding in more depth.
  • Limitation of the study should be described.

Minor issues

  • Text on figure 2 have poor resolution.
  • Graph on figure 3 have poor resolution.
  • Part of figure 3a is cover by label.

Author Response

The authors thank the reviewers for their valuable suggestions and remarks.

We have answered the questions (see below) and modified the manuscript accordingly.

We hope that the modified version of the manuscript is accepted by the editor and the reviewers.

Response to Reviewer 1 Comments:

Point 1: Novelty of the paper is questionable. Study by Zelichenko et al., described SRV-CHX stents for urethra (Zelichenko, Genady, et al. "Prevention of initial biofilm formation on ureteral stents using a sustained releasing varnish containing chlorhexidine: in vitro study." Journal of endourology 27.3 (2013): 333-337.) Moreover, the authors previously described method of SRV-CHX stent production (Gefter Shenderovich, J.; Zaks, B.; Kirmayer, D.; Lavy, E.; Steinberg, D.; Friedman, M., Chlorhexidine sustained-releasevarnishes for catheter coating - Dissolution kinetics and antibiofilm properties. Eur J Pharm Sci 2018, 112, 1-7. ) What is actually novel in your study?

Response 1: The novelty of our study is the use of a sustained-release technology in the nose. Sustained release varnish containing chlorhexidine intended for the sinonasal cavity, that has not yet been described, according to extensive search in the literature. Indeed, sustained release technology has been described in several medical areas but not in the nose. The nasal stents new clinical approach and the use of a sustained release delivery coating may elevate their clinical potential. The nose stents are different solid support from any other support used before for coating.  The support may affect duration of release. Therefore, it was important to study whether the varnish could be kept adherent on the nasal stents and the chlorhexidine released is sufficient to avoid bacterial growth in the surroundings and on the stents.

Abstract properly covers the main points of the study and key words are properly chosen. Title is clear and effective. Authors cited 20 references, majority of them were published within last 10 years. Study design does not rise ethical concerns. Language quality is sufficient

Critical issues

Point 2: Please provide more data on antimicrobial properties of stents. Usage of only two bacterial species is not enough. Please report antibacterial properties of your stents against other pathogens associated with rhinosinusitis.

Response 2: The stents themselves do not show anti-microbial activities, they are used after sinus surgery to keep the neo-ostium open, allowing efficient drainage of sinus secretion. It is the coating with SRV-CHX that exerts this function. Chlorhexidine is a well-known antiseptics that show broad activity to diverse bacteria, including Gram-positive Staphylococcus aureus and the Gram-negative Pseudomonas aeruginosa (doi:10.1021/acsomega.9b00297; doi:10.3390/microorganisms8121991; doi: 10.1007/978-3-319-15126-7_11), that are often involved in rhinosinusitis (Bendouah et al. doi:10.1016/j.otohns.2006.03.001.). Chlorhexidine can also act on preformed biofilms (doi: 10.1007/978-3-319-15126-7_11) and was therefore chosen as the anti-microbial compound in our system.

Point 3: Usage of only two bacterial strains is not enough. It is advised to use more than one strain of each pathogen. Typically, bacterial strains have different susceptibility to xenobiotics.

Response 3: The study is a proof-of-principle that SRV-CHX can be used in the prevention of bacterial growth both on the nose stents and in their vicinities. The two bacterial strains studied represent the two main strains normally involved bacteria in rhinosinusitis. Since CHX is known to act on a wide range of bacterial species, the SRV-CHX coating is expected also to protect against other bacteria. Future studies will focus on the effect of the SRV-CHX-coated stents on the microbiome of the sinonasal cavity. The next study will broaden the tested microbes to others and also test heterogenous biofilm.

Point 4: Authors used only reference strains of bacteria. Lack of assessment of antimicrobial properties of patient-isolated bacteria is another downfall of the paper.

Response 4: We decided to use the most common biofilm-forming bacteria involved in chronic rhinosinusitis (CRS). Information on these bacteria can be found in the EPOS 2020, page 120.  The latter is the European Position Paper on Rhinosinusitis and Nasal Polyps, a guideline for management of this disease that is used by all otolaryngologist all over the world (Fokkens et al. doi:10.4193/Rhin20.600). Multiple bacterial species have been implicated in biofilm formation in CRS including Staphylococcus aureus, Pseudomonas aeruginosa, Haemophilus influenza and Moraxella cattarhalis (Maina et al. doi:10.1007/s40136-018-0212-6). Of these, S. aureus biofilms have the greatest association with severely recurrent and recalcitrant cases of CRS (doi: 10.2500/ajra.2009.23.3413)

We used two well-accepted laboratorial strains of the Gram-positive Staphylococcus aureus and the Gram-negative Pseudomonas aeruginosa for our study. The aim of our study was not to study the effect of CHX on various bacterial strains which has already been extensively documented in the literature, but to prove that a sustained release varnish system of CHX, coating nose stents, can be used in preventing bacterial growth for an extended period of time. Our study shows that this can be achieved with our system. After proven the concept on two representative bacteria – the next study will also test clinical isolated bacteria.

We thank the referee for the two above important suggestions. We have added the above in the discussion, last section.

Point 5: Study lacks any data about cytotoxicity of the stents. It is impossible to analyze antimicrobial properties of stents without knowing their impact on mammalian cells. In my opinion at least simple cytotoxicity screen (e.g. MTT, XTT or almar blue test) should be performed with usage of at least two cell lines derived from nasopharynges.

Response 5: Chlorhexidine has been used as a spray to treat rhinosinusitis and has been shown to be well tolerated. Spray application has the disadvantage that the dose is given in one boost resulting in a temporarily high concentration. In contrast, our slow release system will keep a constant low, but efficient, concentration in the surroundings. As we used the well-studied chlorhexidine that is accepted for medical oral and topical usages for decades and whose cytotoxic activities are well-documented, we didn’t repeat these assays in our study.

Major issues

Point 6: Introduction is vaguely written. Please proved more detail on the diseased as well as current knowledge about SRV stents.

Response 6: We have added more information to the Introduction section.

Point 7: Discussion is way too short. Please discuss your finding in more depth.

Response 7: We have now added text to the discussion.

Point 8: Limitation of the study should be described.

Response 8: We have added limitation to the Discussion.

Minor issues

Point 9: Text on figure 2 have poor resolution.

Response 9: Resolution has now been improved.

Point 10: Graph on figure 3 have poor resolution.

Response 10: Resolution has now been improved.

Point 11: Part of figure 3a is cover by label.

Response 11: Thanks for the notion. This has now been corrected.

Reviewer 2 Report

pharmaceutics-1407905 peer review

This is simple, but interesting work with potential application in prevention of post treatment bacterial contaminations. Authors have performed set of experiments following the logic work plan. In my opinion paper can be recommended to be accepted for publication, however, some corrections, adjustments, and a bit better focus on discussion needs to be performed.

L4: italics for in vitro

L59: Gram needs to be written with capital G, since is referring to Danish scientist Gram.

L70: Provide state name for Hercules Inc. USA. Is this CA?

L71: Since on L69 you have introduced Sigma, then on L71 "PEG400; Sigma" will be sufficient. Do not need to provide full address. Please, check entire manuscript for similar adjustments. Please, check if is "Sigma" or "Sigma Aldrich".

On L46, full names of Staphylococcus aurues and Pseudomonas aeruginosa were introduced, please, in this position and future positions in the paper, apply abbreviated versions for the named species: S. aureus and P. aeruginosa.

Moreover, according to ATCC website, ATCC 25923 is Staphylococcus aureus subsp. aureus, in this context, will be correct to write all around S. aureus subsp. aureus.

Strain PAO1 is belong to what culture collections? Please, specify this information.

L91: Please, write S. aureus subsp. aureus ATCC 25923 or P. aeroginosa PAO1. In entire text, applied pathogens needs to be accompanied by their strains numbers. The perfored test are related to these specific starins and you cannot guaranty that same approach will work in same way against other strains from same species, thus is why strain numbers needs to be provident. Maybe in order to be suggesting this can be applied as really practice approach for control the mentioned pathogens, more representatives strains from same species needs to be simple tested if they will be sensitive to the same antimicrobial composition proposed.

Maybe authors can decide if figure 2 or figure 3 can be used, since both reporting on same experiment, or maybe incorporate photographs (Fig 2) on the graphic (Fig 3)?

Please, correct figure 3 from the block appearing on 3A.

L186: "three drops of each sample...", maybe will be more appropriate if you can specify the volume of that 3 drops. Will be more scientifically correct.

L257: I will suggest to the authors to start discussion with more moderate statement, such as 'On our best of knowledge, this is the first in vitro study..."

L287, 309, 310, etc: italics for in vivo

Maybe if authors can enrich a bit more discussion section and compare their work with some similar studies on different antimicrobials, or different test strains, releases of antimicrobials from coated material, from ninofibers or incorporated to the gels can be good addendum to present study.

In present form discussion is a bit basic, and needs upgrade and update.

Author Response

The authors thank the reviewers for their valuable suggestions and remarks.

We have answered the questions (see below) and modified the manuscript accordingly.

We hope that the modified version of the manuscript is accepted by the editor and the reviewers.

Response to Reviewer 2 Comments:

Point 1: L4: italics for in vitro

Response 1: It is now changed into italics font.

Point 2: L59: Gram needs to be written with capital G, since is referring to Danish scientist Gram.

Response 2: This has been corrected.

Point 3: L70: Provide state name for Hercules Inc. USA. Is this CA?

Response 3: This has been added.

Point 4: L71: Since on L69 you have introduced Sigma, then on L71 "PEG400; Sigma" will be sufficient. Do not need to provide full address. Please, check entire manuscript for similar adjustments. Please, check if is "Sigma" or "Sigma Aldrich".

Response 4: This has been corrected.

Point 5: On L46, full names of Staphylococcus aurues and Pseudomonas aeruginosa were introduced, please, in this position and future positions in the paper, apply abbreviated versions for the named species: S. aureus and P. aeruginosa.

Moreover, according to ATCC website, ATCC 25923 is Staphylococcus aureus subsp. aureus, in this context, will be correct to write all around S. aureus subsp. aureus.

Strain PAO1 is belong to what culture collections? Please, specify this information.

Response 5: This has now been added to the text.

Point 6: L91: Please, write S. aureus subsp. aureus ATCC 25923 or P. aeroginosa PAO1. In entire text, applied pathogens needs to be accompanied by their strains numbers. The perfored test are related to these specific starins and you cannot guaranty that same approach will work in same way against other strains from same species, thus is why strain numbers needs to be provident. Maybe in order to be suggesting this can be applied as really practice approach for control the mentioned pathogens, more representatives strains from same species needs to be simple tested if they will be sensitive to the same antimicrobial composition proposed.

Response 6: We have now added the stain numbers throughout the text.

Point 7: Maybe authors can decide if figure 2 or figure 3 can be used, since both reporting on same experiment, or maybe incorporate photographs (Fig 2) on the graphic (Fig 3)?

Please, correct figure 3 from the block appearing on 3A.

Response 7: We have now fused Figure 2 and 3 into one Figure.

Point 8: L186: "three drops of each sample...", maybe will be more appropriate if you can specify the volume of that 3 drops. Will be more scientifically correct.

Response 8: To test the viability of the bacteria in the samples, three drops of 10 µl from each sample were seeded on agar plates, and the bacterial growth inspected after a 24 h incubation.

Point 9: L257: I will suggest to the authors to start discussion with more moderate statement, such as 'On our best of knowledge, this is the first in vitro study.

Response 9: We have corrected the sentence accordingly: “To our best knowledge, this is the first in vitro study applying chlorhexidine in a sustained release technology as a coating on nasal stents for the prevention of biofilm formation in the nasal cavity.”

Point 10: L287, 309, 310, etc: italics for in vivo

Response 10: It is now changed into italics font.

Point 11: Maybe if authors can enrich a bit more discussion section and compare their work with some similar studies on different antimicrobials, or different test strains, releases of antimicrobials from coated material, from ninofibers or incorporated to the gels can be good addendum to present study.

In present form discussion is a bit basic, and needs upgrade and update.

Response 11: We have now added more information to the Discussion. We have elaborated on: The clinical used of sinonasal stent in the actuality, more on the SRV concept and about the limitations of the study.

Round 2

Reviewer 1 Report

Authors provided some needed changes and explanations. I'm still not convinced in terms on novelty and usage of only two bacterial strains. Although I am not entirely happy with provided changes and I believe quality of paper would benefit if some of experiments which I suggested would be performed, I can agree that paper meet basic criteria for publication. 

Reviewer 2 Report

Dear Editor

I would like to suggest that paper will be accepted for publication.

Only authors will need to correct "subsp." all around the text to NOT Italics.